# INDIVIDUAL PRIVACY ACCOUNTING FOR DIFFERENTIALLY PRIVATE STOCHASTIC GRADIENT DESCENT

## ABSTRACT

Differentially private stochastic gradient descent (DP-SGD) is the workhorse algorithm for recent advances in private deep learning. It provides a single privacy guarantee to all datapoints in the dataset. We propose an efficient algorithm to compute privacy guarantees for individual examples when releasing models trained by DP-SGD. We use our algorithm to investigate individual privacy parameters across a number of datasets. We find that most examples enjoy stronger privacy guarantees than the worst-case bound. We further discover that the training loss and the privacy parameter of an example are well-correlated. This implies groups that are underserved in terms of model utility are simultaneously underserved in terms of privacy guarantee. For example, on CIFAR-10, the average $\varepsilon$ of the class with the lowest test accuracy is 43.6% higher than that of the class with the highest accuracy.

## 1 INTRODUCTION

Differential privacy is a strong notion of data privacy, enabling rich forms of privacy-preserving data analysis (Dwork & Roth, 2014). Informally speaking, it quantitatively bounds the maximum influence of any datapoint using a privacy parameter $\varepsilon$, where a small value of $\varepsilon$ corresponds to stronger privacy guarantees. Training deep models with differential privacy is an active research area (Papernot et al., 2017; Bu et al., 2020; Yu et al., 2022; Anil et al., 2021; Li et al., 2022; Golatkar et al., 2022; Mehta et al., 2022; De et al., 2022; Bu et al., 2022). Models trained with differential privacy not only provide theoretical privacy guarantee to their data but also are more robust against empirical attacks (Bernau et al., 2019; Carlini et al., 2019; Jagielski et al., 2020; Nasr et al., 2021).

Differentially private stochastic gradient descent (DP-SGD) is the de-facto choice for differentially private deep learning (Song et al., 2013; Bassily et al., 2014; Abadi et al., 2016). DP-SGD first clips individual gradients and then adds Gaussian noise to the average of clipped gradients. Standard privacy accounting takes a worst-case approach, and provides all examples with the same privacy parameter $\varepsilon$. However, from the perspective of machine learning, different examples can have very different impacts on a learning algorithm (Koh & Liang, 2017; Feldman & Zhang, 2020). For example, consider support vector machines: removing a non-support vector has no effect on the resulting model, and hence that example would have perfect privacy.

In this paper, we give an efficient algorithm to accurately estimate individual privacy parameters of models trained by DP-SGD. Our privacy guarantee adapts to the training trajectory of one execution of DP-SGD to provide a precise characterization of privacy cost (see Section 2.1 for more details). Inspecting individual privacy parameters allows us to better understand example-wise impacts. It turns out that, for common benchmarks, many examples experience much stronger privacy guarantee than the worst-case bound. To illustrate this, we plot the individual privacy parameters of MNIST (LeCun et al., 1998), CIFAR-10 (Krizhevsky, 2009), and UTKFace (Zhang et al., 2017) in Figure 1. Experimental details, as well as more results, can be found in Section 4 and 5. The disparity in individual privacy guarantees naturally arises when running DP-SGD. To the best of our knowledge, our investigation is the first to explicitly reveal such disparity.

We propose two techniques to make individual privacy accounting viable for DP-SGD. First, we maintain estimates of the gradient norms for all examples so the individual privacy costs can be computed accurately at update. Second, we round the gradient norms with a small precision $r$ to control the number of different privacy costs, which need to be computed numerically. We explain

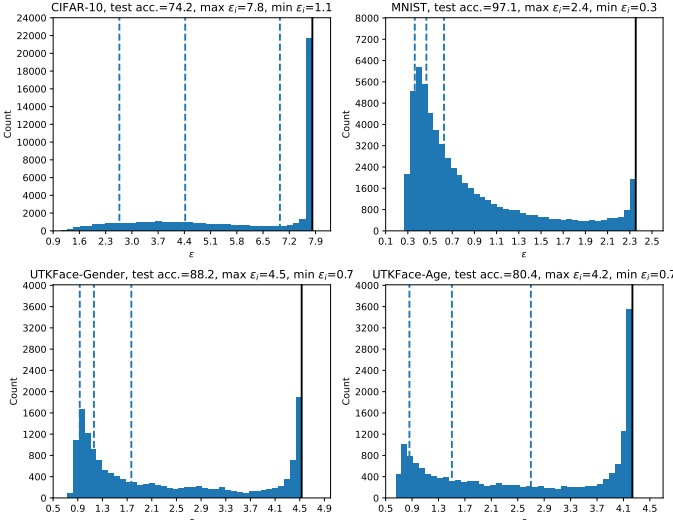

Figure 1: Individual privacy parameters of models trained by DP-SGD. The value of $\delta$ is $1 \times 10^{-5}$. The dashed lines indicate $10\%$, $30\%$, and $50\%$ of datapoints. The black solid line shows the privacy parameter of the original analysis.

why these two techniques are necessary in Section 2. More details of the proposed algorithm, as well as methods to release individual privacy parameters, are in Section 3.

We further demonstrate a strong correlation between the privacy parameter of an example and its final training loss. We find that examples with higher training loss also have higher privacy parameters in general. This suggests that the same examples suffer a simultaneous unfairness in terms of worse privacy and worse utility. While prior works have shown that underrepresented groups experience worse utility (Buolamwini & Gebru, 2018), and that these disparities are amplified when models are trained privately Bagdasaryan et al. (2019); Suriyakumar et al. (2021); Hansen et al. (2022); Noe et al. (2022), we are the first to show that the privacy guarantee *and* utility are negatively impacted concurrently. This is in comparison to prior work in the differentially private setting which took a worst-case perspective for privacy accounting, resulting in a uniform privacy guarantee for all training examples. For instance, when running gender classification on UTKFace, the average $\varepsilon$ of the race with the lowest test accuracy is 25% higher than that of the race with the highest accuracy. We also study the disparity in privacy when models are trained without differential privacy, which may be of independent interest to the community. We use the success rates of membership inference attacks to measure privacy in this case and show groups with worse accuracy suffer from higher privacy risks.

## 1.1 RELATED WORK

Several works have explored example-wise privacy guarantees in differentially private learning. Jorgensen et al. (2015) propose *personalized differential privacy* that provides pre-specified individual privacy parameters which are independent of the learning algorithm, e.g., users can choose different levels of privacy guarantees based on their sensitivities to privacy leakage (Mühl & Boenisch, 2022). A recent line of works also uses the variation in example-wise sensitivities that naturally arise in learning to study example-wise privacy. *Per-instance differential privacy* captures the privacy parameter of a target example with respect to a fixed training set (Wang, 2019; Redberg & Wang, 2021; Golatkar et al., 2022). Feldman & Zrnic (2021) design an individual *Rényi differential privacy filter*. The filter stops when the accumulated cost reaches a target budget that is defined before training. It allows examples with smaller per-step privacy costs to run for more steps. The final privacy guarantee offered by the filter is still the worst-case over all possible outputs as the predefined budget has to be independent of the algorithm outcomes. In this work, we propose output-specific differential privacy and give an efficient algorithm to compute individual guarantees of DP-SGD. We further discover that the disparity in individual privacy parameters correlates well with the disparity in utility.

## 2 PRELIMINARIES

We first give some background on differential privacy. Then we highlight the challenges in computing individual privacy for DP-SGD. Finally, we argue that providing the same privacy bound to all examples is not ideal because of the variation in individual gradient norms.

### 2.1 BACKGROUND ON DIFFERENTIALLY PRIVATE LEARNING

Differential privacy builds on the notion of neighboring datasets. A dataset $\mathbb{D} = \{d_i\}_{i=1}^n$ is a neighboring dataset of $\mathbb{D}'$ (denoted as $\mathbb{D} \sim \mathbb{D}'$) if $\mathbb{D}'$ can be obtained by adding/removing one example from $\mathbb{D}$. The privacy guarantees in this work take the form of $(\varepsilon, \delta)$-differential privacy.

**Definition 1.** *[Individual $(\varepsilon, \delta)$-DP] For a datapoint $d$, let $\mathbb{D}$ be an arbitrary dataset and $\mathbb{D}' = \mathbb{D} \cup \{d\}$ be its neighboring dataset. An algorithm $\mathcal{A}$ satisfies $(\varepsilon(d), \delta)$-individual DP if for any subset of outputs $\mathbb{S} \subset Range(\mathcal{A})$ it holds that*

$$\Pr[\mathcal{A}(\mathbb{D}) \in \mathbb{S}] \le e^{\varepsilon(d)} \Pr[\mathcal{A}(\mathbb{D}') \in \mathbb{S}] + \delta \text{ and } \Pr[\mathcal{A}(\mathbb{D}') \in \mathbb{S}] \le e^{\varepsilon(d)} \Pr[\mathcal{A}(\mathbb{D}) \in \mathbb{S}] + \delta.$$

We further allow the privacy parameter $\varepsilon$ to be a function of a subset of outcomes to provide a sharper characterization of privacy. We term this variant as *output-specific* $(\varepsilon, \delta)$-DP. It shares similar insights as the *ex-post* DP in Ligett et al. (2017); Redberg & Wang (2021) as both definitions tailor the privacy guarantee to algorithm outcomes. Ex-post DP characterizes privacy after observing a particular outcome. In contrast, the canonical $(\varepsilon, \delta)$-DP is an *ex-ante* privacy notion. Ex-ante refers to 'before sampling from the distribution'. Ex-ante DP builds on property about the distribution of the outcome. When the set $\mathbb{A}$ in Definition 2 contains more than one outcome, output-specific $(\varepsilon, \delta)$-DP remains ex-ante because it measures how the outcome is distributed over $\mathbb{A}$.

**Definition 2.** *[Output-specific individual $(\varepsilon, \delta)$-DP] Fix a datapoint $d$ and a set of outcomes $\mathbb{A} \subset Range(\mathcal{A})$, let $\mathbb{D}$ be an arbitrary dataset and $\mathbb{D}' = \mathbb{D} \cup \{d\}$. An algorithm $\mathcal{A}$ satisfies $(\varepsilon(\mathbb{A}, d), \delta)$-output-specific individual DP for $d$ at $\mathbb{A}$ if for any $\mathbb{S} \subset \mathbb{A}$ it holds that*

$$\Pr[\mathcal{A}(\mathbb{D}) \in \mathbb{S}] \le e^{\varepsilon(\mathbb{A}, d)} \Pr[\mathcal{A}(\mathbb{D}') \in \mathbb{S}] + \delta \text{ and } \Pr[\mathcal{A}(\mathbb{D}') \in \mathbb{S}] \le e^{\varepsilon(\mathbb{A}, d)} \Pr[\mathcal{A}(\mathbb{D}) \in \mathbb{S}] + \delta.$$

Definition 2 has the same semantics as Definition 1 once the algorithm's outcome belongs to $\mathbb{A}$ is known. It is a strict generalization of $(\varepsilon, \delta)$-DP as one can recover $(\varepsilon, \delta)$-DP by maximizing $\varepsilon(\mathbb{A}, d)$ over $\mathbb{A}$ and $d$. Making this generalization is crucial for us to precisely compute the privacy parameters of models trained with DP-SGD. The output of DP-SGD is a sequence of models $\{\theta_t\}_{t=1}^T$, where $T$ is number of iterations. The privacy risk at step $t$ highly depends on the training trajectory (the first $t-1$ models). For example, one can adversarially modify the training data and hence change the training trajectory to maximize the privacy risk of a target example (Tramèr et al., 2022). With Definition 2, we can analyze the privacy guarantee of DP-SGD with regards to a fixed training trajectory, which specifies a subset of DP-SGD's outcomes. In comparison, canonical DP analyzes the worst-case privacy over all possible outcomes which could give loose privacy parameters.

A common approach for doing deep learning with differential privacy is to make each update differentially private instead of protecting the trained model directly. The composition property of differential privacy allows us to reason about the overall privacy of running several such steps. We give a simple example to illustrate how to privatize each update. Suppose we take the sum of all gradients $v = \sum_{i=1}^n g_i$ from dataset $\mathbb{D}$. Without loss of generality, further assume we add an arbitrary example $d'$ to obtain a neighboring dataset $\mathbb{D}'$. The summed gradient becomes $v' = v + g'$, where $g'$ is the gradient of $d'$. If we add isotropic Gaussian noise with variance $\sigma^2$, then the output distributions of two neighboring datasets are

$$\mathcal{A}(\mathbb{D}) \sim \mathcal{N}(v, \sigma^2 I) \text{ and } \mathcal{A}(\mathbb{D}') \sim \mathcal{N}(v', \sigma^2 I).$$

We then can bound the divergence between two Gaussian distributions to prove differential privacy, e.g., through *Rényi differential privacy (RDP)* (Mironov, 2017). We give the definition of individual RDP in Appendix A. The expectations of $\mathcal{A}(\mathbb{D})$ and $\mathcal{A}(\mathbb{D}')$ differ by $g'$. A larger gradient leads to a larger difference and hence a worse privacy parameter.

### 2.2 CHALLENGES OF COMPUTING INDIVIDUAL PRIVACY PARAMETERS FOR DP-SGD

Privacy accounting in DP-SGD is more complex than the simple example in Section 2.1 because the analysis involves *privacy amplification by subsampling*. Roughly speaking, randomly sampling

a minibatch in DP-SGD strengthens the privacy guarantees since most points in the dataset are not involved in a single step. How subsampling complicates the theoretical privacy analysis has been studied extensively (Abadi et al., 2016; Balle et al., 2018; Mironov et al., 2019; Zhu & Wang, 2019; Wang et al., 2019). In this section, we focus on how subsampling complicates the empirical computation of individual privacy parameters.

Before we expand on these difficulties, we first describe the output distributions of neighboring datasets in DP-SGD (Abadi et al., 2016). Poisson sampling is assumed, i.e., each example is sampled independently with probability $p$. Let $\boldsymbol{v} = \sum_{i \in \mathbb{M}} \boldsymbol{g}_i$ be the sum of the minibatch of gradients of $\mathbb{D}$, where $\mathbb{M}$ is the set of sampled indices. Consider also a neighboring dataset $\mathbb{D}'$ that has one datapoint with gradient $\boldsymbol{g}'$ added. Because of Poisson sampling, the output is exactly $\boldsymbol{v}$ with probability $1 - p$ ($\boldsymbol{g}'$ is not sampled) and is $\boldsymbol{v}' = \boldsymbol{v} + \boldsymbol{g}'$ with probability $p$ ($\boldsymbol{g}'$ is sampled). Suppose we still add isotropic Gaussian noise, the output distributions of two neighboring datasets are

$$\mathcal{A}(\mathbb{D}) \sim \mathcal{N}(\boldsymbol{v}, \sigma^2 \boldsymbol{I}), \tag{1}$$

$$\mathcal{A}(\mathbb{D}') \sim \mathcal{N}(\boldsymbol{v}, \sigma^2 \boldsymbol{I}) \text{ with prob. } 1 - p, \ \mathcal{A}(\mathbb{D}') \sim \mathcal{N}(\boldsymbol{v}', \sigma^2 \boldsymbol{I}) \text{ with prob. } p. \tag{2}$$

With Equation (1) and (2), we explain the challenges in computing individual privacy parameters.

### 2.2.1 FULL BATCH GRADIENT NORMS ARE REQUIRED AT EVERY ITERATION

There is some privacy cost for $\boldsymbol{d}'$ even if it is not sampled in the current iteration because the analysis makes use of the subsampling process. For a given sampling probability and noise variance, the amount of privacy cost is determined by $\|\boldsymbol{g}'\|$. Therefore, we need accurate gradient norms of all examples to compute accurate privacy costs at every iteration. However, when running SGD, we only compute minibatch gradients. Previous analysis of DP-SGD evades this problem by simply assuming all examples have the maximum possible norm, i.e., the clipping threshold.

### 2.2.2 COMPUTATIONAL COST OF INDIVIDUAL PRIVACY PARAMETERS IS HUGE

The density function of $\mathcal{A}(\mathbb{D}')$ is a mixture of two Gaussian distributions. Abadi et al. (2016) compute the Rényi divergence between $\mathcal{A}(\mathbb{D})$ and $\mathcal{A}(\mathbb{D}')$ numerically to get tight privacy parameters. Although there are some asymptotic bounds, those bounds are looser than numerical computation, and thus such numerical computations are necessary in practice (Abadi et al., 2016; Wang et al., 2019; Mironov et al., 2019; Gopi et al., 2021). In the classic analysis, there is only one numerical computation as all examples have the same privacy cost over all iterations. However, naive computation of individual privacy parameters would require up to $n \times T$ numerical computations, where $n$ is the dataset size and $T$ is the number of iterations.

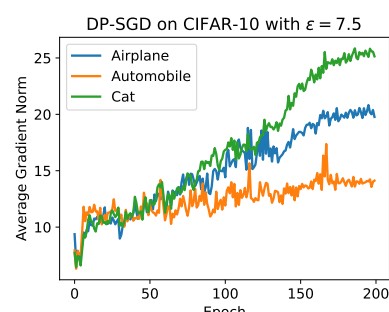

Figure 2: Gradient norms on CIFAR-10.

### 2.3 AN OBSERVATION: GRADIENT NORMS IN DEEP LEARNING VARY SIGNIFICANTLY

As shown in Equation 1 and 2, the privacy parameter of an example is determined by its gradient norm once the noise variance is given. It is worth noting that examples with larger gradient norms usually have higher training loss. This implies that the privacy cost of an example correlates with its training loss, which we empirically demonstrate in Section 4 and 5. In this section, we show gradient norms of different examples vary significantly to demonstrate that different examples experience very different privacy costs. We train a ResNet-20 model with DP-SGD. The maximum clipping threshold is the median of gradient norms at initialization. We plot the average norms of three different classes in Figure 2. The gradient norms of different classes show significant stratification. Such stratification naturally leads to different individual privacy costs. This suggests that quantifying individual privacy parameters may be valuable despite the aforementioned challenges.

## 3 DEEP LEARNING WITH INDIVIDUAL PRIVACY

We give an efficient algorithm (Algorithm 1) to estimate individual privacy parameters for DP-SGD. The privacy analysis of Algorithm 1 is in Section 3.1. We perform two modifications to make individual privacy accounting feasible with small computational overhead. First, we compute full

batch gradient norms once in a while, e.g., at the beginning of every epoch, and use the results to estimate the gradient norms for subsequent iterations. We show the estimations of gradient norms are accurate in Section 3.2. Additionally, we round the gradient norms to a given precision so the number of numerical computations is independent of the dataset size. More details of this modification are in Section 3.3. Finally, we discuss how to make use of the individual privacy parameters in Section 3.4.

In Algorithm 1, we clip individual gradients with estimations of gradient norms (referred to as *individual clipping*). This is different from the vanilla DP-SGD that uses the maximum threshold to clip all examples (Abadi et al., 2016). Individual clipping gives us exact upper bounds on gradient norms and hence we have exact bounds on privacy costs. Our analysis is also applicable to vanilla DP-SGD though the privacy parameters in this case become estimates of exact guarantees, which is inevitable because one can not have exact bounds on gradient norms at every iteration when running vanilla SGD. We report the results for vanilla DP-SGD in Appendix C.2. Our observations include: (1) the estimates of privacy guarantees for vanilla DP-SGD are very close to the exact ones, (2) all our conclusions in the main text still hold when running vanilla DP-SGD, (3) the individual privacy of Algorithm 1 is very close to that of vanilla DP-SGD.

---

**Algorithm 1:** Deep Learning with Individual Privacy Accounting

**Input** : Maximum clipping threshold $C$, rounding precision $r$, noise variance $\sigma^2$, sampling probability $p$, frequency of updating full gradient norms at every epoch $K$, number of epochs $E$.

1 Let $\{C^{(i)}\}_{i=1}^n$ be estimated gradient norms of all examples and initialize $C^{(i)} = C$.
2 Let $\mathbb{C} = \{r, 2r, 3r, \ldots, C\}$ be all possible norms under rounding.
3 **for** $c \in \mathbb{C}$ **do**
4     Compute Rényi divergences at different orders between Equation (1), (2) and store the results in $\boldsymbol{\rho}_c$.
5 **end**
6 Let $\{\boldsymbol{o}^{(i)} = \boldsymbol{0}\}_{i=1}^n$ be the accumulated Rényi divergences at different orders.
7 **for** $e = 1$ *to* $E$ **do**
8     **for** $t' = 1$ *to* $\lfloor 1/p \rfloor$ **do**
9        $t = t' + \lfloor 1/p \rfloor \times (e - 1)$ //*Global step count.*
10        **if** $t \bmod \lfloor 1/pK \rfloor = 0$ **then**
11           Compute full batch gradient norms $\{S^{(i)}\}_{i=1}^n$.
12           Update gradient estimates with rounded norms $\{C^{(i)} = \arg\min_{c \in \mathbb{C}}(|c - S^{(i)}|)\}_{i=1}^n$.
13        **end**
14        Sample a minibatch of gradients $\{\boldsymbol{g}^{(I_j)}\}_{j=1}^{|I|}$, where $I$ is the sampled indices.
15        Clip gradients $\bar{\boldsymbol{g}}^{(I_j)} = clip(\boldsymbol{g}^{(I_j)}, C^{(I_j)})$.
16        Update model $\theta_t = \theta_{t-1} - \eta(\sum \bar{\boldsymbol{g}}^{(I_j)} + z)$, where $z \sim \mathcal{N}(0, \sigma^2 \boldsymbol{I})$.
17        **for** $i = 1$ *to* $n$ **do**
18           Set $\boldsymbol{\rho}^{(i)} = \boldsymbol{\rho}_c$ where $c = C^{(i)}$.
19           $\boldsymbol{o}^{(i)} = \boldsymbol{o}^{(i)} + \boldsymbol{\rho}^{(i)}$. //*Update individual privacy costs.*
20        **end**
21     **end**
22 **end**

---

## 3.1 Privacy Analysis of Algorithm 1

We state the privacy guarantee of Algorithm 1 in Theorem 3.1. We use output-specific DP to precisely account the privacy costs of a realized run of Algorithm 1.

**Theorem 3.1.** *[Output-specific privacy guarantee] Algorithm 1 at step $t$ satisfies $(o_\alpha^{(i)} + \frac{\log(1/\delta)}{\alpha-1}, \delta)$-output-specific individual DP for the $i_{th}$ example at $\mathbb{A} = (\theta_1, \ldots, \theta_{t-1}, \Theta_t)$, where $o_\alpha^{(i)}$ is the accumulated RDP at order $\alpha$ and $\Theta_t$ is the domain of $\theta_t$.*

To prove Theorem 3.1, we first define a $t$-step non-adaptive composition with $\theta_1, \ldots, \theta_{t-1}$. We then show RDP of the non-adaptive composition gives an output-specific privacy bound on the adaptive composition in Algorithm 1. We relegate the proof to Appendix A.

## 3.2 Estimations of Gradient Norms Are Accurate

Although the gradient norms used for privacy accounting are updated only occasionally, we show that the individual privacy parameters based on estimations are very close to the those based on exact

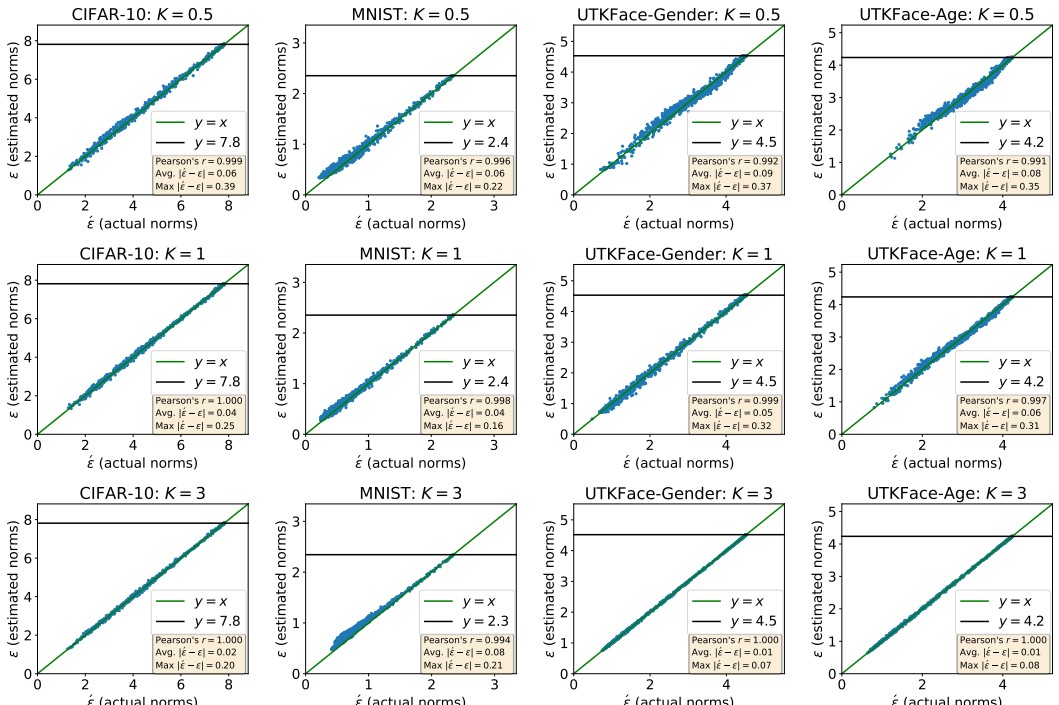

Figure 3: Privacy parameters based on estimations of gradient norms ($\varepsilon$) versus those based on exact norms ($\acute{\varepsilon}$). The results suggest that the estimations of gradient norms are close to the exact norms.

norms (Figure 3). This indicates that the estimations of gradient norms are close to the exact ones. We comment that the privacy parameters computed with exact norms are not equivalent to those of vanilla DP-SGD because individual clipping in Algorithm 1 slightly modifies the training trajectory. We show individual clipping has a small influence on the individual privacy of vanilla DP-SGD in Appendix C.2.

To compute the exact gradient norms, we randomly sample 1000 examples and compute the exact gradient norms at every iteration. We compute the Pearson correlation coefficient between the privacy parameters computed with estimated norms and those computed with exact norms. We also compute the average and the worst absolute errors. We report results on MNIST, CIFAR-10, and UTKFace. Details about the experiments are in Section 4. We plot the results in Figure 3. The $\varepsilon$ values based on estimations are close to those based on exact norms (Pearson's $r > 0.99$) even we only update the gradient norms every two epochs ($K = 0.5$). Updating full batch gradient norms more frequently further improves the estimation, though doing so would increase the computational overhead.

It is worth noting that the maximum clipping threshold $C$ affects the computed privacy parameters. Large $C$ increases the variation of gradient norms (and hence the variation of privacy parameters) but leads to large noise variance while small $C$ suppresses the variation and leads to large gradient bias. Large noise variance and gradient bias are both harmful for learning (Chen et al., 2020; Song et al., 2021). In Appendix D, we show the influence of using different $C$ on both accuracy and privacy.

### 3.3 ROUNDING INDIVIDUAL GRADIENT NORMS

The rounding operation in Algorithm 1 is essential to make the computation of individual privacy parameters feasible. The privacy cost of one example at each step is the Rényi divergences between Equation (1) and (2). For a fixed sampling probability and noise variance, we use $\rho_c$ to denote the privacy cost of an example with gradient norm $c$. Note that $\rho_c$ is different for every value of $c$. Consequently, there are at most $n \times E$ different privacy costs because individual gradient norms vary across different examples and epochs ($n$ is the number of datapoints and $E$ is the number of training epochs). In order to make the number of different $\rho_c$ tractable, we round individual gradient norms with a prespecified precision $r$ (see Line 12 in Algorithm 1). Because the maximum clipping

threshold $C$ is usually a small constant, then, by the pigeonhole principle, there are at most $\lceil C/r \rceil$ different values of $\boldsymbol{\rho}_c$. Throughout this paper we set $r = 0.01C$ that has almost no impact on the precision of privacy accounting.

Table 1: Computational costs of computing individual privacy parameters for CIFAR-10 with $K = 1$.

|  | With Rounding | Without Rounding |
|---|---|---|
| # of numerical computations | $1 \times 10^2$ | $1 \times 10^7$ |
| Time (in seconds) | $< 3$ | $\sim 2.6 \times 10^4$ |

We give a concrete comparison on the computational costs in Table 1. We run the numerical method in Mironov et al. (2019) once for every different privacy cost (with the default setup in the Opacus library (Yousefpour et al., 2021)). We run DP-SGD on CIFAR-10 for 200 epochs with $K = 1$. All results in Table 1 use multiprocessing with 5 cores of an AMD EPYC™ 7V13 CPU. With rounding, the overhead of computing individual privacy parameters is negligible. In contrast, the computational cost without rounding is more than 7 hours.

### 3.4 WHAT CAN WE DO WITH INDIVIDUAL PRIVACY PARAMETERS?

Note that individual privacy parameters are dependent on the private data and thus sensitive, and consequently may not be released publicly without care. We describe some approaches to safely make use of individual privacy parameters.

The first approach is to release $\varepsilon_i$ to the owner of the $i$th example. Although we use gradient norms without adding noise, this approach does not incur additional privacy loss for two reasons. First, it is safe for the $i$th example because only the rightful owner sees $\varepsilon_i$. Second, releasing $\varepsilon_i$ does not increase the privacy loss of any other examples. This is because the computation only uses $(\theta_1, \ldots, \theta_{t-1})$, which is reported in a privacy-preserving manner. We proof the claim in Appendix E.1.

The second approach is to privately release aggregate statistics of the population, e.g., the average or quantiles of the $\varepsilon$ values. Recent works have demonstrated such statistics can be published accurately with minor privacy cost Andrew et al. (2021). We show the statistics can be released accurately with a very small privacy parameter ($\varepsilon = 0.1$) in Appendix E.2.

Finally, individual privacy parameters can also serve as a powerful tool for a trusted data curator to improve the model quality. By analysing the individual privacy parameters of a dataset, a trusted curator can focus on collecting more data representative of the groups that have higher privacy risks to mitigate the disparity in privacy.

### 4 INDIVIDUAL PRIVACY PARAMETERS ON DIFFERENT DATASETS

We first show the distribution of individual privacy parameters of running DP-SGD on four classification tasks in Section 4.1. Then we investigate how individual privacy parameters correlate with training loss in Section 4.2. Experimental setup is as follows.

**Datasets.** We use two benchmark datasets MNIST ($n = 60000$) and CIFAR-10 ($n = 50000$) (LeCun et al., 1998; Krizhevsky, 2009) as well as the UTKFace dataset ($n \simeq 15000$) (Zhang et al., 2017) that contains the face images of four different races (White, $n \simeq 7000$; Black, $n \simeq 3500$; Asian, $n \simeq 2000$; Indian, $n \simeq 2800$). We construct two tasks on UTKFace: predicting gender, and predicting whether the age is under 30.[1] We slightly modify the dataset between these two tasks by randomly removing a few examples to ensure each race has balanced positive and negative labels.

**Models and hyperparameters.** For CIFAR-10, we use the WRN16-4 model in De et al. (2022), which achieves advanced performance in private setting. We follow the implementation details in De et al. (2022) expect their data augmentation method to reduce computational cost. For MNIST and UTKFace, we use ResNet20 models with batch normalization layers replaced by group normalization layers. For UTKFace, we initialize the model with weights pre-trained on ImageNet. We set $C = 1$ on CIFAR-10, following De et al. (2022). For MNIST and UTKFace, we set $C$ as the median of

---

[1] We acknowledge that predicting gender and age from images may be problematic. Nonetheless, as facial images have previously been highlighted as a setting where machine learning has disparate accuracy on different groups, we revisit this domain through a related lens. The labels are provided by the dataset curators.

Figure 4: Privacy parameters and final training losses. Each point shows the loss and privacy parameter of one example. Pearson's $r$ is computed between privacy parameters and log losses.

gradient norms at initialization, following the suggestion in Abadi et al. (2016). We set $K = 3$ for all experiments in this section. More details about the hyperparameters are in Appendix B.

### 4.1 INDIVIDUAL PRIVACY PARAMETERS VARY SIGNIFICANTLY

Figure 1 shows the individual privacy parameters on all datasets. The privacy parameters vary across a large range on all tasks. On the CIFAR-10 dataset, the maximum $\varepsilon_i$ value is 7.8 while the minimum $\varepsilon_i$ value is only 1.1. We also observe that, for easier tasks, more examples enjoy stronger privacy guarantees. For example, $\sim$40% of examples reach the worst-case $\varepsilon$ on CIFAR-10 while only $\sim$3% do so on MNIST. This may because the loss decreases quickly when the task is easy, resulting in gradient norms also decreasing and thus a reduced privacy loss.

### 4.2 PRIVACY PARAMETERS AND LOSS ARE POSITIVELY CORRELATED

We study how individual privacy parameters correlate with individual training loss. The analysis in Section 2 suggests that the privacy parameter of one example depends on its gradient norms across training. Intuitively, an example would have high loss after training if its gradient norms are large. We visualize individual privacy parameters and the final training loss values in Figure 4. The individual privacy parameters on all datasets increase with loss until they reach the maximum $\varepsilon$. To quantify the order of correlation, we further fit the points with one-dimensional logarithmic functions and compute the Pearson correlation coefficients with the privacy parameters predicted by the fitted curves. The Pearson correlation coefficients are larger than $0.9$ on all datasets, showing an logarithmic correlation between the privacy parameter of a datapoint and its training loss.

## 5 GROUPS ARE SIMULTANEOUSLY UNDERSERVED IN BOTH ACCURACY AND PRIVACY

### 5.1 LOW-ACCURACY GROUPS HAVE WORSE PRIVACY PARAMETERS

It is well-documented that machine learning models may have large differences in accuracy on different groups (Buolamwini & Gebru, 2018; Bagdasaryan et al., 2019). Our finding demonstrates that this disparity may be simultaneous in terms of both accuracy *and* privacy. We empirically verify this by plotting the average $\varepsilon$ and training/test accuracy of different groups. The experiment setup is the same as Section 4. For CIFAR-10 and MNIST, the groups are the data from different classes, while for UTKFace, the groups are the data from different races.

We plot the results in Figure 5. The groups are sorted based on the average $\varepsilon$. Both training and test accuracy correlate well with $\varepsilon$. Groups with worse accuracy do have worse privacy guarantee in general. On CIFAR-10, the average $\varepsilon$ of the 'Cat' class (which has the worst test accuracy) is 43.6% higher than the average $\varepsilon$ of the 'Automobile' class (which has the highest test accuracy). On UTKFace-Gender, the average $\varepsilon$ of the group with the lowest test accuracy ('Asian') is 25.0% higher than the average $\varepsilon$ of the group with the highest accuracy ('Indian'). Similar observation also holds on other tasks. To the best of our knowledge, our work is the first to reveal this simultaneous disparity.

### 5.2 LOW-ACCURACY GROUPS SUFFER FROM HIGHER EMPIRICAL PRIVACY RISKS

We run membership inference (MI) attacks against models trained **without** differential privacy to study whether underserved groups are more vulnerable to empirical privacy attacks. Several recent works show that MI attacks have higher success rates on some examples than other examples (Song

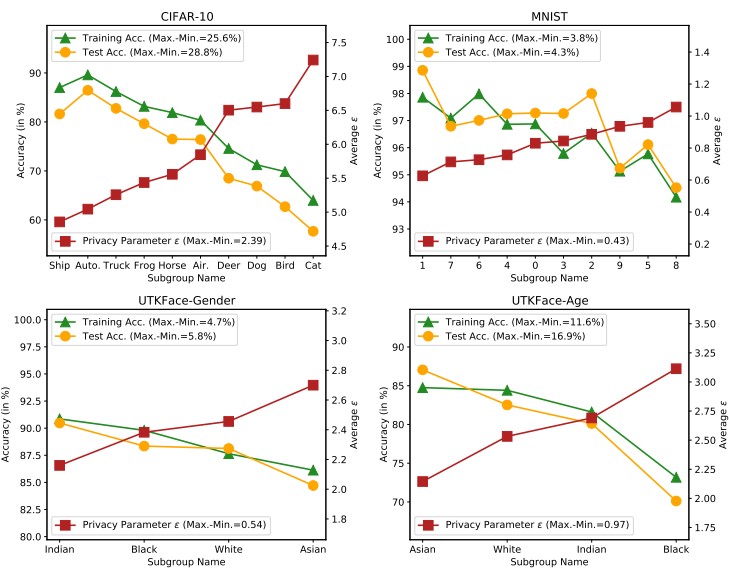

Figure 5: Accuracy and average $\varepsilon$ of different groups. Groups with worse accuracy also have worse privacy in general.

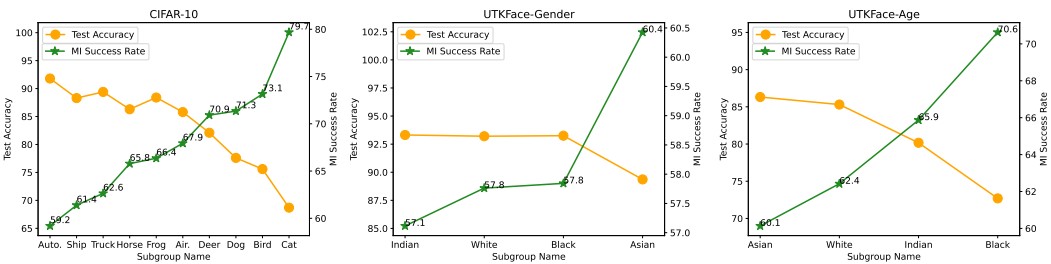

Figure 6: MI attack success rates on different groups. Target models are trained without DP

& Mittal, 2020; Choquette-Choo et al., 2021; Carlini et al., 2021). In this section, we directly connect such disparity in privacy risks with the unfairness in utility. We use a simple loss-threshold attack that predicts an example is a member if its loss value is smaller than a prespecified threshold (Sablayrolles et al., 2019). For each group, we use its whole test set and a random subset of training set so the numbers of training and test losses are balanced. We further split the data into two subsets evenly to find the optimal threshold on one and report the success rate on another. More implementation details are in Appendix B. The results are in Figure 6. The groups are sorted based on their average $\varepsilon$. The disparity in privacy risks is clear. On UTKFace-Age, the MI success rate is 70.6% on the 'Black' group while is only 60.1% on the 'Asian' group. This suggests that empirical privacy risks also correlate well with the disparity in utility.

## 6 CONCLUSION

We propose an efficient algorithm to accurately estimate the individual privacy parameters for DP-SGD. We use this new algorithm to examine individual privacy guarantees on several datasets. Significantly, we find that groups with worse utility also suffer from worse privacy. This new finding reveals the complex while interesting relation among utility, privacy, and fairness. It has two immediate implications. Firstly, it shows that the learning objective aligns with privacy protection for a given privacy budget, i.e., the better the model learns about an example, the better privacy that example would get. Secondly, it suggests that mitigating the utility fairness under differential privacy is more tricky than doing so in the non-private case. This is because classic methods such as upweighting underserved examples would exacerbate the disparity in privacy.

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

# A  PROOF OF THEOREM 3.1

**Theorem 3.1.** *[Output-specific privacy guarantee] Algorithm 1 at step $t$ satisfies $(o_\alpha^{(i)} + \frac{\log(1/\delta)}{\alpha-1}, \delta)$-output-specific individual DP for the $i_{th}$ example at $\mathbb{A} = (\theta_1, \ldots, \theta_{t-1}, \Theta_t)$, where $o_\alpha^{(i)}$ is the accumulated RDP at order $\alpha$ and $\Theta_t$ is the domain of $\theta_t$.*

Here we give the the proof of Theorem 3.1. We compose the privacy costs at different steps through Rényi differential privacy (RDP) (Mironov, 2017). RDP uses the Rényi divergence at different orders to measure privacy. We use $D_\alpha(\mu||\nu) = \frac{1}{\alpha-1} \log \int (\frac{d\mu}{d\nu})^\alpha d\nu$ to denote the Rényi divergence at order $\alpha$ between $\mu$ and $\nu$ and $D_\alpha^{\leftrightarrow}(\mu||\nu) = \max(D_\alpha(\mu||\nu), D_\alpha(\nu||\mu))$ to denote the maximum divergence of two directions. The definition of individual RDP is as follows.

**Definition 3.** *[Individual RDP Feldman & Zrnic (2021)] Fix a datapoint $\boldsymbol{d}$, let $\mathbb{D}$ be an arbitrary dataset and $\mathbb{D}' = \mathbb{D} \cup \{\boldsymbol{d}\}$. A randomized algorithm $\mathcal{A}$ satisfies $(\alpha, \rho(\boldsymbol{d}))$-individual RDP for $\boldsymbol{d}$ if it holds that $D_\alpha^{\leftrightarrow}(\mathcal{A}(\mathbb{D})||\mathcal{A}(\mathbb{D}')) \leq \rho(\boldsymbol{d})$.*

Let $(\mathcal{A}_1, \ldots, \mathcal{A}_{t-1})$ be a sequence of randomized algorithms and $(\theta_1, \ldots, \theta_{t-1})$ be some arbitrary fixed outcomes from the domain, we define

$$\hat{\mathcal{A}}^{(t)}(\theta_1, \ldots, \theta_{t-1}, \mathbb{D}) = (\mathcal{A}_1(\mathbb{D}), \mathcal{A}_2(\theta_1, \mathbb{D}), \ldots, \mathcal{A}_t(\theta_1, \ldots, \theta_{t-1}, \mathbb{D})).$$

Noting that $\hat{\mathcal{A}}^{(t)}$ is not an adaptive composition as the input of each individual mechanism does not depend on the outputs of previous mechanisms. Further let

$$\mathcal{A}^{(t)}(\mathbb{D}) = (\mathcal{A}_1(\mathbb{D}), \mathcal{A}_2(\mathcal{A}_1(\mathbb{D}), \mathbb{D}), \ldots, \mathcal{A}_t(\mathcal{A}_1(\mathbb{D}), \ldots, \mathbb{D}))$$

be the adaptive composition. In Theorem A.1, we show a RDP bound on the non-adaptive composition $\hat{\mathcal{A}}^{(t)}$ gives an output-specific DP bound on the adaptive composition $\mathcal{A}^{(t)}$.

**Theorem A.1.** *Let $\mathbb{A} = (\theta_1, \ldots, \theta_{t-1}, Range(\mathcal{A}_t)) \subset Range(\mathcal{A}^{(t)}) = Range(\hat{\mathcal{A}}^{(t)})$ where $\theta_1, \ldots, \theta_{t-1}$ are some arbitrary fixed outcomes. If $\hat{\mathcal{A}}^{(t)}(\cdot)$ satisfies $o_\alpha$ RDP at order $\alpha$, then $\mathcal{A}^{(t)}(\mathbb{D})$ satisfies $(o_\alpha + \frac{\log(1/\delta)}{\alpha-1}, \delta)$-output-specific differential privacy at $\mathbb{A}$.*

*Proof.* For a given outcome $\theta^{(t)} = (\theta_1, \theta_2, \ldots, \theta_{t-1}, \theta_t) \in \mathbb{A}$, we have $\mathbb{P}\left[\mathcal{A}^{(t)}(\mathbb{D}) = \theta^{(t)}\right] =$

$$\mathbb{P}\left[\mathcal{A}^{(t-1)}(\mathbb{D}) = \theta^{(t-1)}\right] \mathbb{P}\left[\mathcal{A}_t(\mathcal{A}_1(\mathbb{D}), \ldots, \mathcal{A}_{t-1}(\mathbb{D}), \mathbb{D}) = \theta_t | \mathcal{A}^{(t-1)}(\mathbb{D}) = \theta^{(t-1)}\right], \quad (3)$$

$$= \mathbb{P}\left[\mathcal{A}^{(t-1)}(\mathbb{D}) = \theta^{(t-1)}\right] \mathbb{P}\left[\mathcal{A}_t(\theta_1, \ldots, \theta_{t-1}, \mathbb{D}) = \theta_t\right], \quad (4)$$

by the product rule of conditional probability. Apply the product rule recurrently on $\mathbb{P}\left[\mathcal{A}^{(t-1)}(\mathbb{D}) = \theta^{(t-1)}\right]$, we have $\mathbb{P}\left[\mathcal{A}^{(t)}(\mathbb{D}) = \theta^{(t)}\right] =$

$$\mathbb{P}\left[\mathcal{A}^{(t-2)}(\mathbb{D}) = \theta^{(t-2)}\right] \mathbb{P}\left[\mathcal{A}_{t-1}(\theta_1, \ldots, \theta_{t-2}, \mathbb{D}) = \theta_{t-1}\right] \mathbb{P}\left[\mathcal{A}_t(\theta_1, \ldots, \theta_{t-1}, \mathbb{D}) = \theta_t\right], \quad (5)$$

$$= \mathbb{P}\left[\mathcal{A}_1(\mathbb{D}) = \theta_1\right] \mathbb{P}\left[\mathcal{A}_2(\theta_1, \mathbb{D}) = \theta_2\right] \ldots \mathbb{P}\left[\mathcal{A}_t(\theta_1, \ldots, \theta_{t-1}, \mathbb{D}) = \theta_t\right], \quad (6)$$

$$= \mathbb{P}\left[\hat{\mathcal{A}}^{(t)}(\theta_1, \ldots, \theta_{t-1}, \mathbb{D}) = \theta^{(t)}\right]. \quad (7)$$

In words, $\mathcal{A}^{(t)}$ and $\hat{\mathcal{A}}^{(t)}$ are identical in $\mathbb{A}$. Therefore, $\mathcal{A}^{(t)}$ satisfies $(\varepsilon, \delta)$-DP at any $\mathbb{S} \subset \mathbb{A}$ if $\hat{\mathcal{A}}^{(t)}$ satisfies $(\varepsilon, \delta)$-DP. Converting the RDP bound on $\hat{\mathcal{A}}^{(t)}(\mathbb{D})$ into a $(\varepsilon, \delta)$-DP bound with Lemma A.2 then completes the proof.

**Lemma A.2** (Conversion from RDP to $(\varepsilon, \delta)$-DP Mironov (2017)). *If $\mathcal{A}$ satisfies $(\alpha, \rho)$-RDP, then $\mathcal{A}$ satisfies $(\rho + \frac{\log(1/\delta)}{\alpha-1}, \delta)$-DP for all $0 < \delta < 1$.*

$\square$

We comment that the RDP parameters of the adaptive composition of Algorithm 1 are random variables because they depend on the randomness of the intermediate outcomes. This is different from the conventional analysis that chooses a constant privacy parameter before training. Composition of random RDP parameters requires additional care because one needs to upper bound the privacy parameter over its randomness (Feldman & Zrnic, 2021; Lécuyer, 2021; Whitehouse et al., 2022). In this paper, we focus on the realizations of those random RDP parameters and hence provide a precise output-specific privacy bound.

## B  MORE IMPLEMENTATION DETAILS

The batchsize is 4K, 2K, and 200 for CIFAR-10, MNIST, and UTKFace, respectively. The training epoch is 300 for CIFAR-10 and 100 for MNIST and UTKFace. We use the package in Yousefpour et al. (2021) to compute the noise multiplier. The standard deviation of noise in Algorithm 1 is the noise multiplier times the maximum clipping threshold. The non-private models in Section 5.2 are as follows. For CIFAR-10, we use a ResNet20 model in He et al. (2016) that has $\sim$0.2M parameters, with all batch normalization layers replaced by group normalization layers. For UTKFace, we use the same models in Section 4. We remove both gradient clipping and noise for non-private training. Other settings are the same as those in Section 4. All experiments are run on a single Tesla V100 GPU with 32G memory. Our source code, including the implementation of our algorithm as well as scripts to reproduce the plots, will be released soon.

## C  MORE DISCUSSION ON INDIVIDUAL CLIPPING

### C.1  INDIVIDUAL CLIPPING DOES NOT AFFECT ACCURACY

Here we run experiments to check the influence of using individual clipping thresholds on utility. Algorithm 1 uses individual clipping thresholds to ensure the computed privacy parameters are strict privacy guarantees. If the clipping thresholds are close to the actual gradient norms, then the clipped results are close to those of using a single maximum clipping threshold. However, if the estimations of gradient norms are not accurate, individual thresholds would clip more signal than using a single maximum threshold.

Table 2: Comparison between the test accuracy of using individual clipping thresholds and that of using a single maximum clipping threshold. The maximum $\varepsilon$ is 7.8 for CIFAR-10 and 2.4 for MNIST.

|  | CIFAR-10 | MNIST |
|---|---|---|
| Individual | 74.0 ($\pm$0.19) | 97.17 ($\pm$0.12) |
| Maximum | 74.1 ($\pm$0.24) | 97.26 ($\pm$0.11) |

We compare the accuracy of two different clipping methods in Table 2. The individual clipping thresholds are updated once per epoch. We repeat the experiment four times with different random seeds. Other setups are the same as those in Section 4. The results suggest that using individual clipping thresholds in Algorithm 1 has a negligible effect on accuracy.

### C.2  EXPERIMENTS WITHOUT INDIVIDUAL CLIPPING

We run Algorithm 1 without individual clipping, i.e., vanilla DP-SGD in Abadi et al. (2016), to see whether our conclusions in the main text still hold. Specifically, we change the Line 15 of Algorithm 1 to clip all gradients with the maximum clipping threshold $C$. Other experimental setup is the same as that in Experiment 4 and Appendix B.

**We get accurate estimations of actual guarantees.**    The privacy parameters are still computed with the estimations of gradient norms and hence are estimations of the actual guarantees. We compare the privacy parameters and actual guarantees in Figure 7. To compute the actual guarantees, we randomly sample 1000 examples and compute their exact gradient norms at every iteration. The results suggest that the privacy parameters are accurate estimations (Pearson's $r > 0.99$ and small maximum absolute errors).

**Privacy parameters still have a strong correlation with training loss.**    In Figure 8, we show privacy parameters computed without individual clipping are still positively correlated with training losses. The Pearson correlation coefficient between privacy parameters and log losses is larger than

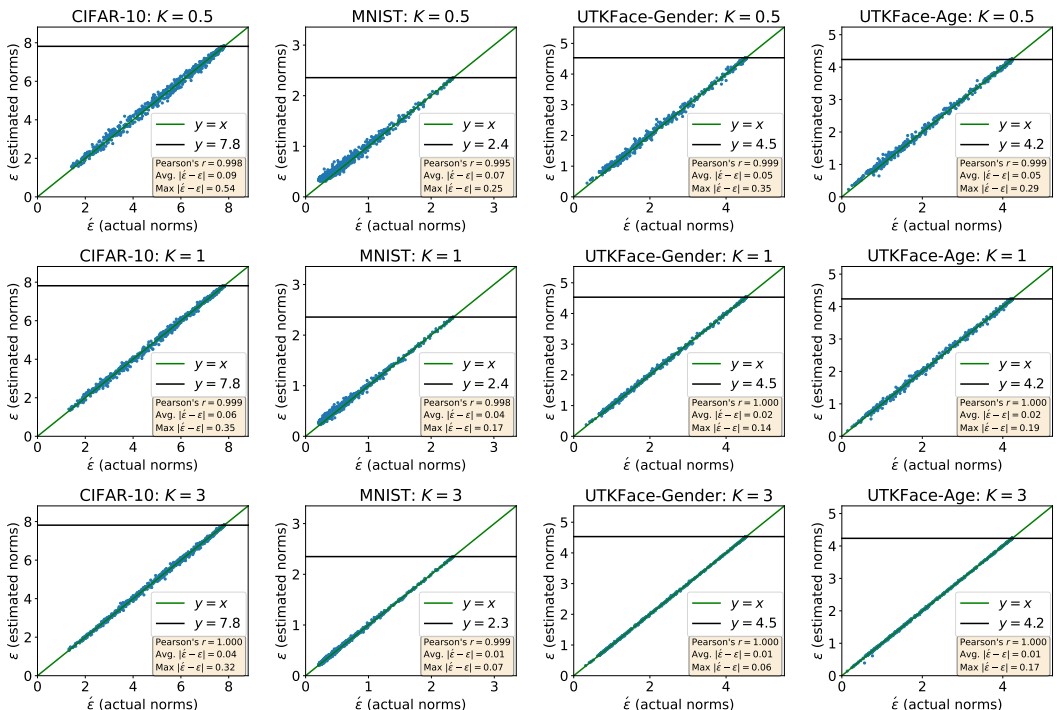

Figure 7: Privacy parameters based on estimations of gradient norms ($\varepsilon$) versus the actual privacy guarantees ($\acute{\varepsilon}$). We do not use individual clipping in this plot. The privacy parameters are very close to the actual guarantees.

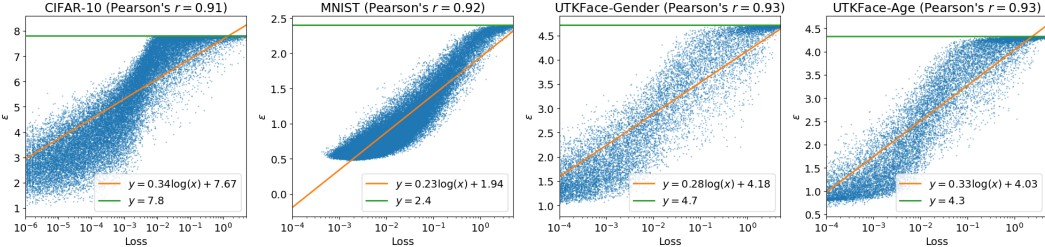

Figure 8: Privacy parameters and final training losses. The experiments are run without individual clipping. The Pearson correlation coefficient is computed between privacy parameters and log losses.

0.9 for all datasets. Moreover, we show our observation in Section 5, i.e., low-accuracy groups have worse privacy parameters, still holds in Figure 9. We also make a direct comparison with privacy parameters computed with individual clipping. We find that privacy parameters computed with individual clipping are very close to those computed without individual clipping. We also find that the order of groups, sorted by the average $\varepsilon$, is exactly the same for both cases.

## D  THE INFLUENCE OF MAXIMUM CLIPPING ON INDIVIDUAL PRIVACY

The value of the maximum clipping threshold $C$ would affect individual privacy parameters. A large value of $C$ would increase the stratification in gradient norms but also increase the noise variance for a fixed privacy budget. A small value of $C$ would suppress the stratification but also increase the gradient bias. Here we run experiments with different values of $C$ on CIFAR-10. We use a ResNet20 for CIFAR-10 in He et al. (2016), which only has $\sim$0.2M parameters, to reduce the computation cost. All batch normalization layers are replaced with group normalization layers. Let $M$ be the median of gradient norms at initialization, we choose $C$ from the list $[0.1M, 0.3M, 0.5M, 1.5M]$. Other experimental setup is the same as that in Section 4.

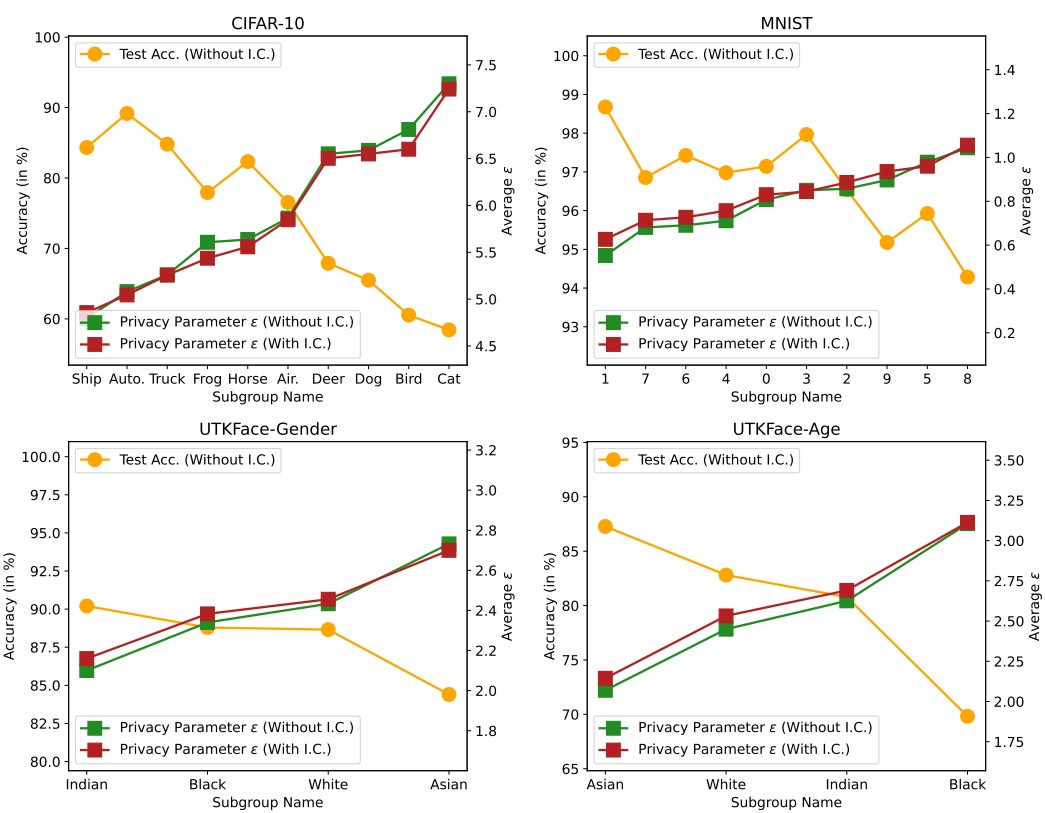

Figure 9: Test accuracy and privacy parameters computed with/without individual clipping (I.C.). Groups with worse test accuracy also have worse privacy in general.

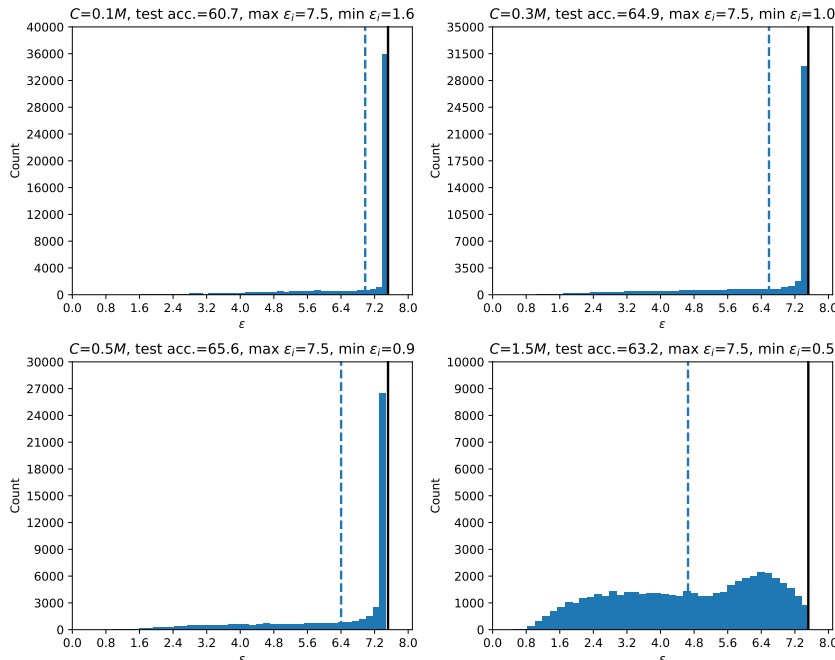

Figure 10: Distributions of individual privacy parameters on CIFAR-10 with different maximum clipping thresholds. The dashed line indicates the average of privacy parameters.

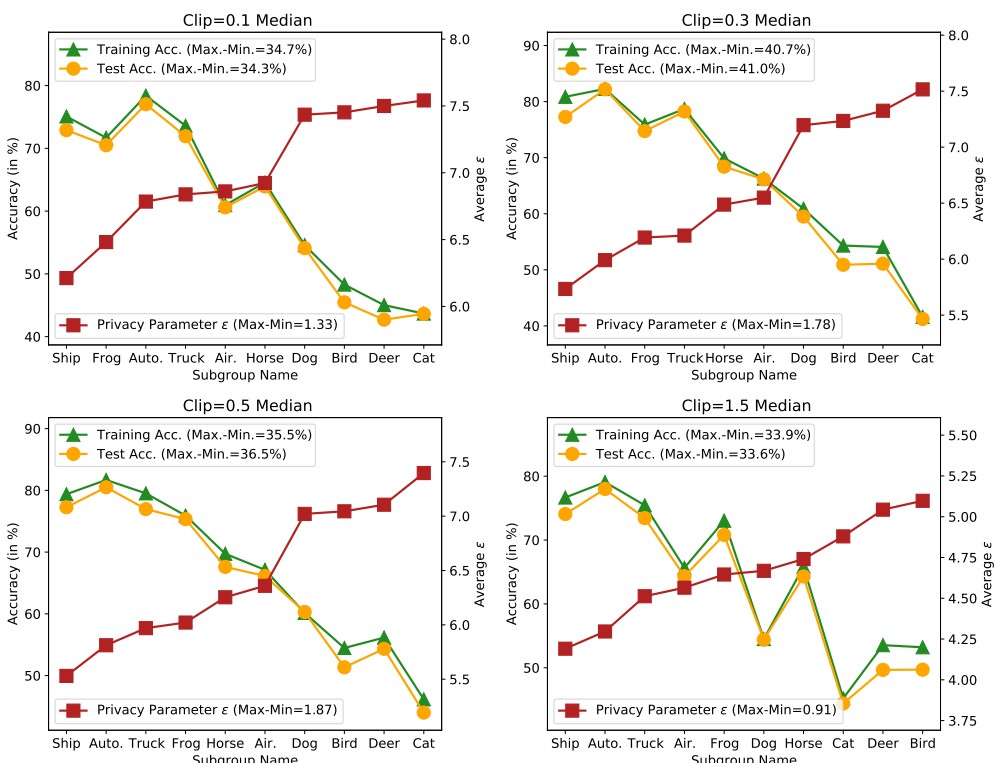

Figure 11: Accuracy and average $\varepsilon$ of different groups on CIFAR-10 with different maximum clipping thresholds.

The histograms of individual privacy parameters are in Figure 10. In terms of accuracy, using clipping thresholds near the median gives better test accuracy. In terms of privacy, using smaller clipping thresholds increases privacy parameters in general. The number of datapoints that reaches the worst privacy decreases with the value of $C$. When $C = 0.1M$, nearly 70% datapoints reach the worst privacy parameter while only $\sim$2% datapoints reach the worst parameter when $C = 1.5M$.

The correlation between accuracy and privacy is in Figure 11. The disparity in average $\varepsilon$ is clear for all choices of $C$. Another important observation is that when decreasing $C$, the privacy parameters of underserved groups increase quicker than other groups. When changing $C = 1.5M$ to $0.5M$, the average $\varepsilon$ of 'Cat' increases from 4.8 to 7.4, almost reaching the worst-case bound. In comparison, the increment in $\varepsilon$ of the 'Ship' class is only 1.3 (from 4.2 to 5.5).

## E    MAKE USE OF INDIVIDUAL PRIVACY PARAMETERS

### E.1    RELEASING INDIVIDUAL PRIVACY PARAMETERS TO RIGHTFUL OWNERS

Let $\varepsilon_i$ be the privacy parameter of the $i_{th}$ user, we can compute $\varepsilon_i$ with the training trajectory and $\boldsymbol{d}_i$ itself. Theorem E.1 shows that releasing $\varepsilon_i$ does not increase the privacy cost of any other example $\boldsymbol{d}_j \neq \boldsymbol{d}_i$. The proof uses the fact that computing $\varepsilon_i$ can be seen as a post-processing of $(\theta_1, \ldots, \theta_{t-1})$, which is reported in a privacy-preserving manner.

**Theorem E.1.** *Let $\mathcal{A} : \mathcal{D} \rightarrow \mathcal{O}$ be an algorithm that is $(\varepsilon_j, \delta)$-output-specific DP for $\boldsymbol{d}_j$ at $\mathbb{A} \subset \mathcal{O}$. Let $f(\cdot, \boldsymbol{d}_i) : \mathcal{O} \rightarrow \mathcal{R} \times \mathcal{O}$ be a post-processing function that returns the privacy parameter of $\boldsymbol{d}_i$ ($\neq \boldsymbol{d}_j$) and the training trajectory. We have $f$ is $(\varepsilon_j, \delta)$-output-specific DP for $\boldsymbol{d}_j$ at $\mathbb{F} \subset \mathcal{R} \times \mathcal{O}$ where $\mathbb{F} = \{f(a, \boldsymbol{d}_i) : a \in \mathbb{A}\}$ is all possible post-processing results.*

*Proof.* We first note that the construction of $f$ is independent of $\boldsymbol{d}_j$. Without loss of generality, let $\mathbb{D}, \mathbb{D}' \in \mathcal{D}$ be the neighboring datasets where $\mathbb{D}' = \mathbb{D} \cup \{\boldsymbol{d}_j\}$. Let $\mathbb{S} \subset \mathbb{F}$ be an arbitrary event and

Table 3: Populational statistics of individual privacy parameters on MNIST. The average estimation error rate is 1.19%. The value of $\delta$ is $1 \times 10^{-5}$.

| MNIST | Average | 0.1-quantile | 0.3-quantile | Median | 0.7-quantile | 0.9-quantile |
|---|---|---|---|---|---|---|
| Non-private | 0.850 | 0.362 | 0.467 | 0.626 | 0.931 | 1.840 |
| $\varepsilon = 0.1$ | 0.852 | 0.361 | 0.469 | 0.628 | 0.935 | 1.742 |

Table 4: Populational statistics of individual privacy parameters on CIFAR-10. The average estimation error rate is 1.51%. The value of $\delta$ is $1 \times 10^{-5}$.

| CIFAR-10 | Average | 0.1-quantile | 0.3-quantile | Median | 0.7-quantile | 0.9-quantile |
|---|---|---|---|---|---|---|
| Non-private | 5.813 | 2.870 | 4.957 | 6.602 | 7.290 | 7.488 |
| $\varepsilon = 0.1$ | 5.811 | 2.960 | 4.975 | 6.609 | 7.357 | 7.828 |

$\mathbb{T} = \{a \in \mathbb{A} : f(a, \boldsymbol{d}_i) \in \mathbb{S}\}$. Because $f$ is a bijective function, we have

$$\Pr\left[f\left(\mathcal{A}(D), \boldsymbol{d}_i\right) \in \mathbb{S}\right] = \Pr\left[\mathcal{A}(D) \in \mathbb{T}\right] \tag{8}$$
$$\leq e^{\varepsilon_j} \Pr\left[\mathcal{A}(D') \in \mathbb{T}\right] + \delta \tag{9}$$
$$= e^{\varepsilon_j} \Pr\left[f\left(\mathcal{A}(D'), \boldsymbol{d}_i\right) \in \mathbb{S}\right] + \delta, \tag{10}$$

which completes the proof.

$\square$

### E.2 Release Populational Statistics of Privacy Parameters

The individual privacy parameters computed by Algorithm 1 are sensitive and hence can not be directly released to the public. Here we show the populational statistics of individual parameters can be released with minor privacy cost. Specifically, we compute the average and quantiles of the $\varepsilon$ values with differential privacy. For average, we release the noisy aggregation through Gaussian Mechanism. For quantiles, we solve the objective function in Andrew et al. (2021) with 20 steps of full batch gradient descent. The results on MNIST and CIFAR-10 are in Table 3 and Table 4 respectively. The released statistics are close to the actual values on both datasets with $(0.1, 10^{-5})$-DP.

