# OpenReview forum: "Individual Privacy Accounting for Differentially Private Stochastic Gradient Descent"
_ICLR.cc/2023/Conference — Submitted to ICLR 2023_

### Official Review · Reviewer_9tdY · 2022-10-22

**Confidence:** 4
**Correctness:** 3
**Technical Novelty And Significance:** 4
**Empirical Novelty And Significance:** 3
**Recommendation:** 6

**Clarity, Quality, Novelty And Reproducibility:**



Figure 2 is not very surprising given that the groups are created based on gradient norm at some point.  It may be more interesting to study the difference in gradient norm across groups constructed in advance not considering any gradient norm.

The steps 3 and 4 in Algorithm 1 are unclear to me, e.g., how exactly does rounding happen, what does "compute RDP" in line 4 mean exactly, where are the results of the "compute RDP" operation in line 4 stored (and where are they used later on), ... ?  The same holds for the explanation in Section 3.3 of the rounding.  It may help to define variables rather than vaguely describe concepts in words such as "different privacy costs" (the privacy costs of what exactly is meant here?)

**Strength And Weaknesses:**


The idea is interesting, it allows to evaluate to what extent an individual suffered privacy loss due to the publication of a learnt model.

The paper could be made clearer (avoiding ambiguity among others by using symbols in addition to text when confusion is possible) and more self-contained (reviewing briefly already existing concepts).



**Summary Of The Paper:**


The paper studies "individual differential privacy", i.e., given a specific example x, to what extent can one determine the membership (in a dataset D) of that example x given the output of a given algorithm?



**Summary Of The Review:**


The paper is interesting and with some more effort can be made sufficiently clear.

---

> ### Author Response · Authors · 2022-11-15
> **Response to Reviewer 9tdY**
>
>
> We thank the reviewer for the suggestions. We are glad that the reviewer likes our analysis on individual privacy of DP-SGD. DP-SGD is the most widely used algorithm for training deep learning models. However, all previous work offers the same privacy analysis for the entire dataset. We believe analyzing the privacy loss from the perspective of individual users is of great interest to the community. Below is our point-to-point response to the suggestions.
>
> **Weakness 1**: The paper could be made clearer and more self-contained.
>
> **Reply**: Thank you for the suggestions. We have revised Algorithm 1 and Section 3.3 to improve the clarity. In Section 2.1, we also add a reference to Appendix A where we give the definitions of most concepts in this paper.  Changes are highlighted in blue color. Our revisions include,
>
> 1. We replace ‘compute RDP’ with ‘compute Rényi divergences at different orders between Equation (1) and (2)’. We also mention the results are stored in $\rho_{c}$. (Line 4 of Algorithm 1)
>
> 2. We use an equation to express the rounding operation. (Line 12 of Algorithm 1)
>
> 3. We use an equation to replace our description of how we find the per-step privacy cost of an example. (Line 18 of Algorithm 1)
>
> 4. In Section 3.3, we clarify the definition of privacy cost. We also express the cost of an example with norm $c$ as $\rho_{c}$.
>
>
>
> **Weakness 2**: “Figure 2 is not very surprising given that the groups are created based on gradient norm at some point. It may be more interesting to study the difference in gradient norm across groups constructed in advance not considering any gradient norm.”
>
> **Reply**: Thank you for the comment. We have changed the plot to show the gradient norms of three classes in CIFAR-10 (‘Airplane’, ‘Automobile’, ‘Cat’). The observation remains the same, i.e., gradient norms of different classes show significant stratification.

---

> ### Comment · Reviewer_9tdY · 2022-12-12
> **further thoughts**
>
>
> After discussion with other reviewers, my opinion changes in the following way:
> * The theoretical work is a bit incremental, even though the slightly different way of presenting the ideas compared to earlier work seems interesting to me.
> * I hence lowered my score to 6 (weak accept): this is a sound paper but maybe it doesn't feature the most substantial advance over existing work.
> * I feel the authors could have better exploited their (incremental) new ideas, e.g., by providing examples of scenarios where their new notions would fit better while the earlier notions would have not so well.  I can imagine such scenarios can exist, but it is better to see them also clearly in the paper as supporting motivation

---

> > ### Author Response · Authors · 2022-12-12
> > **Author response to further thoughts**
> >
> >
> > We thank the reviewer for the new comments. Here we explain why ex-post DP in Ligett el al., (2017) and Redberg and Wang (2021) is not suitable for presenting the results in Section 4 and 5. Ex-post DP is not applicable to approximate DP, e.g., $(\varepsilon,\delta)$-DP that is the default choice in differentially private deep learning. This is because ex-post DP is for a single outcome while $(\varepsilon,\delta)$-DP measures the property of a distribution before sampling from it. In contrast, our output-specific DP can be easily extended to $(\varepsilon,\delta)$-DP  because it  specifies a set of outcomes (a distribution).

---

### Official Review · Reviewer_JYd4 · 2022-10-24

**Confidence:** 5
**Correctness:** 1
**Technical Novelty And Significance:** 1
**Empirical Novelty And Significance:** 1
**Recommendation:** 3

**Clarity, Quality, Novelty And Reproducibility:**

The clarity and quality of the paper are low.




**Strength And Weaknesses:**

Strength: Compared with worst-case DP guarantees, the individual DP guarantee of a sample allows us to better understand the privacy guarantee of a specific data sample, and its impact over model parameters.

Weaknesses:

W1. The proposed algorithm does not correctly estimate individual-level privacy as in Definitions 1 and 2. Hence, the claimed contribution is not justified and the impact of this paper is questionable.

(1) The proposed algorithm modifies the underlying DPSGD algorithm for training. In particular, the algorithm modifies clipping norms for individuals, whereas the original DPSGD clips all gradient samples using the same constant. This change would affect the privacy analysis in DPSGD, and hence, it is unclear whether the algorithm could still correctly estimate the individual privacy guarantee.

(2) The authors claim that their techniques are used for the output-specific individual privacy of DPSGD. Hence, we expect the algorithm to take the output of DPSGD as the input, which is the noisy gradient sum.  However, in the proposed algorithm, only the DP estimates for individual clipping norms (which, again, is not a component of the original DPSGD) are taken into account. It is unclear how the noisy gradient sums in multiple iterations, as the ``trajectory`` of output, impact the output of the algorithm.

(3) No ground truth about individual privacy is provided. Hence, it is confusing how the paper evaluates the accuracy/correctness of the proposed algorithm.

W2. The proposed techniques are straightforward.

(1) The claim that subsampling in DPSGD complicates the privacy analysis is not well justified. The worst-case privacy analysis of DPSGD already incorporates subsampling, as shown in the following papers: "Rényi Differential Privacy of the Sampled Gaussian Mechanism" by Zhang et al., and "Subsampled Rényi Differential Privacy and Analytical Moments Accountant" by Wang et al. Thus, it is why subsampling brings a challenge in the analysis of individual DP.

(2) Overall, the algorithm simply computes the Rényi divergence based on different clipping norms, which is merely an application of existing analysis.

(3) The computation cost of N (which is the size of the private dataset) does not seem to be a problem for individual privacy accounting by definition, since otherwise you would miss some data samples and the accounting is not individual-level anymore (instead it is group level).

(4) The technique for reducing the computation cost is to simply group data samples by their estimates of clipping norms at each iteration. This technique is straightforward.

**Summary Of The Paper:**

This paper presents an algorithm to accurately account for individual differential privacy guarantees in DPSGD.

**Summary Of The Review:**

This paper provides an algorithm for individual privacy analysis in DPSGD. However, the proposed algorithm does not correctly analyze the individual privacy as defined in their paper. In addition, the techniques proposed are straightforward. Furthermore, no ground truth is given.

---

> ### Author Response · Authors · 2022-11-15
> **Response to Reviewer JYd4 (Part 2/2)**
>
> **Weakness 2**: “The proposed techniques are straightforward.”
>
> (1)”The claim that subsampling in DPSGD complicates the privacy analysis is not well justified. The worst-case privacy analysis of DPSGD already incorporates subsampling.”
>
> (2)”Overall, the algorithm simply computes the Rényi divergence based on different clipping norms, which is merely an application of existing analysis.”
>
> (3)”The computation cost of N (which is the size of the private dataset) does not seem to be a problem for individual privacy accounting by definition, since otherwise you would miss some data samples and the accounting is not individual-level anymore (instead it is group level).”
>
> (4)”The technique for reducing the computation cost is to simply group data samples by their estimates of clipping norms at each iteration. This technique is straightforward.”
>
>
> **Reply**: (1) (2): Our claim is that subsampling complicates the **computation** of individual privacy parameters. When subsampling is not used, the privacy parameter has a close-formed solution and one can get it with a single calculation.  However, with subsampling we need to run the tool in Mironov et al., (2019) that numerically computes the Rényi divergence between mixed Gaussian distributions. Such computational costs are high if the number of privacy parameters is large. How subsampling complicates the theoretical analysis of privacy has been studied extensively in the past and we cite relevant papers in our submission. We have revised our submission to make our focus more clear. Specifically, we write “How subsampling complicates the theoretical privacy analysis has been studied extensively. In this section, we focus on how subsampling complicates the empirical computation of individual privacy parameters.”
>
>
> (3) (4): We are glad to see that the reviewer agrees that the number of computations scales with N if one computes the individual privacy parameters straightforwardly.  While our techniques are not complicated, they allow us to efficiently compute individual privacy parameters of the whole dataset for the first time.
>
>
> Finally, we want to emphasize two of our contributions that the reviewer may have overlooked.
>
> 1. We propose a new privacy notion, output-specific $(\varepsilon,\delta)$-DP. This is a necessary preliminary for computing the individual privacy parameters.
>
> 2. We are the first to show that the privacy guarantee of one example has a strong correlation with its training loss, which is a new and nice contribution agreed by other reviewers. Prior to our work, all algorithms using DP-SGD report a uniform privacy guarantee that is a coarse characterization of the exact guarantee. We also discover a novel connection between the disparity in privacy and the unfairness in utility.

---

> ### Author Response · Authors · 2022-11-15
> **Response to Reviewer JYd4 (Part 1/2)**
>
> We thank the reviewer for the detailed comments. We add new experiment results to our revision based on the reviewer’s concerns. All notable changes are highlighted in blue color. Please find our point-to-point response below.
>
>
> **Weakness 1**: “The proposed algorithm does not correctly estimate individual-level privacy as in Definitions 1 and 2. Hence, the claimed contribution is not justified and the impact of this paper is questionable.”
>
> (1)”The proposed algorithm modifies the underlying DPSGD algorithm for training. In particular, the algorithm modifies clipping norms for individuals, whereas the original DPSGD clips all gradient samples using the same constant. This change would affect the privacy analysis in DPSGD, and hence, it is unclear whether the algorithm could still correctly estimate the individual privacy guarantee.”
>
> (2)”The authors claim that their techniques are used for the output-specific individual privacy of DPSGD. Hence, we expect the algorithm to take the output of DPSGD as the input, which is the noisy gradient sum. However, in the proposed algorithm, only the DP estimates for individual clipping norms (which, again, is not a component of the original DPSGD) are taken into account. It is unclear how the noisy gradient sums in multiple iterations, as the trajectory of output, impact the output of the algorithm.”
>
> (3)”No ground truth about individual privacy is provided. Hence, it is confusing how the paper evaluates the accuracy/correctness of the proposed algorithm.”
>
>
> **Reply**: Definitions 1 and 2 are applicable to all algorithms, including DP-SGD and its variants, and we compute the individual-level privacy of Algorithm 1 correctly as per definitions. We believe the reviewer’s concern is that Algorithm 1 is different from the standard DP-SGD because of individual clipping. If this is the reviewer’s concern, we acknowledge that. To address this concern, we run a new set of experiments **without individual clipping**, i.e., using exactly the original DP-SGD. The results are in Appendix C.2. We make a direct comparison of the individual privacy in these two cases (Figure 9). We find that the privacy parameters computed without individual clipping are very close to those computed with. Moreover, the order of groups sorted by average $\varepsilon$ are exactly the same for both cases. We also show the correlation between privacy and loss is still very strong in Figure 8.
>
> Here we explain why we use individual clipping in the main text. When running the original DP-SGD, our privacy accountant does an estimation of individual privacy. This is because, by the nature of SGD, one can not have exact bounds of individual gradient norms at every iteration. Computing privacy parameters with estimates of norms inevitably gives estimates of privacy. Although in Figure 7 we show the estimates of individual privacy for the vanilla DP-SGD are very close to the exact ones (Pearson’s $r>0.99$ and small absolute errors), we opt for doing individual clipping in the main text to get an exact privacy accountant. To compute the exact individual privacy when running the original DP-SGD, we compute the exact gradient norms of 1000 random samples at every iteration.
>
>
> We have revised our submission to include the following changes.
>
> 1. We add a separate paragraph in Section 3 to explain why we use individual clipping.
>
> 2. We revise Section 3.2 to clarify that when individual clipping is applied, privacy parameters computed with exact norms are not equivalent to the privacy parameters of the original DP-SGD because individual clipping may change the training trajectory.
>
> 3. We add the new experiments to Appendix C.2. to demonstrate 1) the individual privacy of Algorithm 1 is very close to that of the original DP-SGD, 2) all our conclusions in the main text still hold when running without individual clipping.

---

> > ### Author Response · Authors · 2022-11-21
> > **More experiments to demonstrate individual clipping has little effect on individual privacy.**
> >
> >
> > Dear Reviewer JYd4,
> >
> > In addition to the group-wise $\varepsilon$ comparison in Figure 9, we provide more experiments to demonstrate the individual privacy of Algorithm 1 is close to that of the original DP-SGD.  We give a point-wise comparison of the privacy parameters in these two cases. Specifically, to compute the ground truth privacy parameters, we still run the original DP-SGD and compute the exact gradient norms of 1000 random samples at every iteration. We compare the privacy parameters returned by Algorithm 1 with the ground truth ones in [this image](https://github.com/AnonymousFigure-Submission164/FigureUpload/blob/main/compare_with_no_ic_fixseed.pdf) (we did get enough time to include this experiment before the revision DDL). When $K\geq 1$, the **Pearson's correlation coefficient is larger than $0.99$** . We fix the random seed in this experiment to ensure that individual clipping is the only factor that affects the training trajectory. We believe this experiment further demonstrates that the individual privacy of Algorithm 1 is very close to that of the original DP-SGD.

---

> ### Author Response · Authors · 2022-11-18
> **Does our response address your concerns?**
>
>
> Dear Reviewer JYd4,
>
> Thanks again for your detailed feedback. We have posted our response to your concerns. Please consider re-evaluate our submission (especially the Correctness part). We would be happy to further discuss. If the reviewer has further suggestions, we would also appreciate hearing them.

---

### Official Review · Reviewer_rPUG · 2022-10-25

**Confidence:** 3
**Correctness:** 3
**Technical Novelty And Significance:** 2
**Empirical Novelty And Significance:** 4
**Recommendation:** 6

**Clarity, Quality, Novelty And Reproducibility:**

This paper is well-written. The statements in the paper are clear and concise. The novelty of this paper is good (see the strengths). I believe that the experimental results are reproducible (also see arXiv 2209.15596).

**Strength And Weaknesses:**

Major strengths:
1. This is the first result showing the correlation between individual training loss and individual privacy parameters. Although this idea is natural if we consider the influence of the data points in the tail of the data distribution on both loss and privacy guarantee, it is great to see such a detailed experimental analysis.

Minor strengths:
1. This paper is easy to follow.
2. The visualization of the experimental results is good.

Major weaknesses:
1. I see that the authors are aware of the work by Feldman and Zenic (2021) on the Renyi Filter. It is not explained in this paper why they choose to use the concept by Redberg and Wang (2021) instead. (The authors are not giving enough acknowledgment to Redberg and Wang (2021) in their Definitions 1 and 2.)
2. There is no theoretical result on the underlying reason behind this correlation between training loss and privacy parameters.
3. In Section 6, the correlation between the attack success rate and the average $\varepsilon$ is from the non-private model. This does not support the relationship between the individual privacy parameters and the actual privacy risk. This part of the experiment needs to be removed or replaced.

Minor weaknesses:
1. In Figure 4, I do not understand why the Pearson correlation coefficient is not calculated with the original points but with the fitted curve.
2. Algorithm 1 needs to be revised. I think $q$ should be $p$ which is the sampling ratio for each batch. The authors have not defined $I_j$, and I guess $m$ is the batch size.
3. In Section 3.4, the authors propose to release $\varepsilon_i$ to the owner of the $i$th example. I think there needs a theoretical proof of why this does not incur additional privacy loss. I think the calculation of $\varepsilon_i$ depends on the differentially private model which depends on the other examples. Therefore, we should be careful about what has been released and how much of the privacy budget has been used up.
4. The usage of individual clipping is discussed in Section C but I do not think this is a common practice. If we do not adopt this individual clipping but only global clipping, will the result still show the correlation between loss and privacy as strong as it is now?



**Summary Of The Paper:**

The authors provide an efficient algorithm for individual privacy accounting when using DP-SGD. By checking the individual privacy parameters, the authors find that these parameters are highly correlated to individual training loss. The authors also verify the validity of their individual privacy parameters by the results of membership inference attacks.

**Summary Of The Review:**

This paper is novel in terms of finding a correlation between training loss and individual privacy. Although without any theoretical analysis of this phenomenon, this paper provides various experimental results to support their finding.

---
 After discussion with Area Chair and other reviewers

---
It would be great if the authors could explain directly how Definition 2 is used in considering the privacy guarantee for a group. Currently, there is a lack of evidence why Definition 2 and Theorem 3.1 is necessary for Section 5, which is the highlight of this paper.

---

> ### Author Response · Authors · 2022-11-15
> **Response to Reviewer rPUG (Part 2/2)**
>
>
> **Minor weaknesses**:
>
> 1.“In Figure 4, I do not understand why the Pearson correlation coefficient is not calculated with the original points but with the fitted curve.”
>
> We want to show the logarithmic curve represents the relation between privacy and training loss. This is equivalent to computing the coefficient with the log loss values and privacy parameters because predicting log loss with the fitted curve is a linear transformation. We have clarified this in the updated draft.
>
> 2.“Algorithm 1 needs to be revised.”
>
> Thanks for pointing these out. We have revised accordingly.
>
> 3.“There needs a theoretical proof of why releasing $\varepsilon_i$ to the owner of the $i$th example does not incur additional privacy loss.”
>
> The calculation of $\varepsilon_i$ depends on other examples but only through $(\theta_{1},\ldots,\theta_{t})$, which is reported in a privacy-preserving manner. We give the proof in Appendix E.1. The main idea is as follows. We first define a bijective function that takes the training trajectory and $d_{i}$ as input and outputs $\varepsilon_{i}$ and the training trajectory. We then show post-processing with the bijective function does not increase the privacy costs of other examples.
>
> 4.”The usage of individual clipping is discussed in Section C but I do not think this is a common practice.”
>
>
>
>
> We add experiments with global clipping. The results are in Appendix C.2. In Figure 8, we show the correlation between privacy and loss is as strong as that in Figure 4. In Figure 9, we show groups are still simultaneously underserved in accuracy and privacy. We also make a direct comparison between the privacy parameters computed with individual clipping and those computed without. We find that the privacy parameters in these two cases are very close. Moreover, the underserved groups are exactly the same for both cases.
>
>
> Although doing global clipping does not modify the vanilla DP-SGD, the privacy accountant in this case only does an estimation of individual privacy.  This is because, by the nature of SGD, we do not have the exact individual gradient norms at every iteration. Computing privacy parameters with estimates of norms inevitably gives estimates of privacy. Therefore, we opt for doing individual clipping in the main text to get an exact privacy accountant. Nonetheless, we show the estimates of privacy guarantees for the vanilla DP-SGD are very close to the exact ones in Figure 7 (Pearson’s $r>0.99$ and small absolute errors). We compute the exact gradient norms of 1000 random samples at every iteration to compute the exact individual privacy.
>
>
> We have revised our submission to 1) add a separate paragraph to explain why we use individual clipping (Section 3), 2) show our observations in the main text still hold when running the vanilla DP-SGD (Appendix C.2).

---

> > ### Comment · Reviewer_rPUG · 2022-11-17
> > **Feedbacks**
> >
> > 1. Got it. I guess you can redraw Figure 4 to have a log scale of the x-axis.
> >
> > 2. Algorithm 1 looks good to me now.
> >
> > 3. This sounds reasonable to me.
> >
> > 4. Thanks for the additional experiments!

---

> > > ### Author Response · Authors · 2022-11-18
> > > **New version of Figure 4**
> > >
> > >
> > > Thanks for the suggestion, we have revised Figure 4 and 8 to have a log scale of the x-axis.

---

> ### Author Response · Authors · 2022-11-15
> **Response to Reviewer rPUG (Part1/2)**
>
> We thank the reviewer for the detailed comments. We have revised our submission accordingly to improve our submission. All notable changes are highlighted in blue color. Below is our point-to-point response.
>
> **Weakness 1**: “It is not explained in this paper why they choose to use the concept by Redberg and Wang (2021) instead.” && “The authors are not giving enough acknowledgement to Redberg and Wang (2021).”
>
> **Reply**: Thanks for the suggestion. We have updated our draft to include 1) why we do not use the privacy filter in Feldman and Zenic (2021); 2) more discussion on the difference between Def. 2 and ex-post DP. We briefly list the discussion as follows.
>
> The privacy filter in Feldman and Zenic (2021) requires fixing a target privacy budget before training. It stops when the accumulated privacy cost reaches the predefined budget. The filter allows examples with smaller per-step privacy costs (i.e., smaller gradient norms in the case of DP learning) to run for more steps. The final privacy guarantee offered by the filter is still the worst-case guarantee, as the overall budget has to be independent of the algorithm outcomes. Such a definition does not reveal the difference of the individual privacy of different samples, which is the main target of this paper. Therefore, we do not use the concept in Feldman and Zenic (2021).
>
>
> Our output-specific DP (Def. 2) follows the insights in Ligett el al., (2017) and Redberg and Wang (2021) as our definition also tailors the privacy guarantee to algorithm outcomes. One difference is that Def. 2 is ex-ante because it gives the guarantee before the final outcome is sampled, while ex-post DP analyzes the privacy of a sampled outcome.
>
>
> **Weakness 2**: “There is no theoretical result on the underlying reason behind this correlation between training loss and privacy parameters.”
>
> **Reply**: Given a fixed noise variance, the privacy parameters depend on gradient norms, which further correlate with training loss. Although examples with higher training loss generally have larger gradient norms, it is hard to theoretically connect the training loss with the per-sample gradient norm in the nonconvex optimization. Therefore, we focus on revealing their empirical correlation as our main scope is to study the individual privacy offered by DP-SGD, which is mostly used to train neural networks.
>
>
> **Weakness 3**: “In Section 6, the correlation between the attack success rate and the average $\varepsilon$  is from the non-private model.”
>
> **Reply**: Thank the reviewer for pointing out this question. Our intention was to demonstrate that the disparity in privacy exists even if models are trained without DP. This is a natural follow-up after we reveal the disparity in privacy for models trained with DP. We believe connecting the disparity in privacy risks with the unfairness in utility for standardly trained deep models will also be  interesting  to the community. To make this point clearer, we have revised the narrative and changed the plot to only show the MI success rates correlate with the test accuracy of models trained without DP.

---

> > ### Comment · Reviewer_rPUG · 2022-11-17
> > **About the 'ex-ante' DP and the attack success rate**
> >
> > Thanks for the detailed response!
> >
> > In your reply to Weakness 1, you mentioned that 'Def. 2 is ex-ante because it gives the guarantee before the final outcome is sampled'. I am not sure why the definition by Ligett el al., (2017) is called 'ex-post,' while your Def. 2 is ex-ante, since the only difference is that Def. 2 is on a subset of outcome while Ligett el al., (2017) consider every single outcome (a single element can be considered as a special case of a subset if you choose the subset to contain only this element). Please explain in detail the difference between 'ex-post' and 'ex-ante' that can be used to distinguish the two definitions.
> >
> > Your reply to Weakness 2 sounds intuitive which also explains the connection between Section 2.3 and the main topic of this paper (the correlation between accuracy loss and privacy loss). However, in your current version of the manuscript, one may still find it hard to understand why Section 2.3 is necessary for the paper. It might be helpful to put your reply in your paper.
> >
> > Your reply to Weakness 3 and the update of Figure 6 look good to me except for one issue: I find that the Test accuracy in Figure 6 is lower than the test ACC in Figure 5, while I guess Figure 5 is training with noise injected but Figure 6 is clean training. This may be questionable since I would expect that clean training usually provides better test accuracy.

---

> > > ### Author Response · Authors · 2022-11-17
> > > **Continued Discussion**
> > >
> > >
> > > Thank you for the quick response! We greatly appreciate the constructive feedback.
> > >
> > >
> > > **1.** Explain the difference between ‘ex-post’ and ‘ex-ante’ in detail.
> > >
> > >
> > > **Reply**: Ex-ante refers to ‘before sampling from the distribution’. In this work, ex-ante DP refers to privacy notions that build on property about the distribution of the outcome. For instance, the canonical $(\varepsilon,\delta)$-DP is ex-ante because it measures the probability/density of arbitrary events over the whole sample space. In contrast, ex-post DP is computed with the probability/density of a particular outcome. Our output-specific $(\varepsilon,\delta)$-DP remains ex-ante because it still measures how the outcome is distributed over a subset of the sample space. We have revised the discussion before Definition 2 to better distinguish these two definitions.
> > >
> > >
> > > **2.** Use reply to Weakness 2 to explain the connection between Section 2.3 and the main topic of this paper.
> > >
> > >
> > > **Reply**: We have revised Section 2.3 to include the connection between gradient norm and privacy cost. We also mention such a connection implies the correlation between accuracy loss and privacy loss. Specifically, we wrote: “As shown in Equation 1 and 2, the privacy parameter of an example is determined by its gradient norm once the noise variance is given. It is worth noting that examples with larger gradient norms usually have higher training loss. This implies that the privacy cost of an example correlates with its training loss, which we empirically demonstrate in Section 4.”.
> > >
> > > **3.** The Test accuracy in Figure 6 may be questionable since I would expect that clean training usually provides better test accuracy.
> > >
> > > **Reply**: We accidentally reported the test accuracy of three DP models in the last revision. The test accuracy of CIFAR-10 was from a ResNet20 model that has ~0.2M parameters. The test accuracy of UTKFace was from the models in Section 5.1. Figure 6 has been revised to report the test accuracy of corresponding non-DP models. The correlation between test accuracy and MI attack success rates does not change. We apologize for this confusion. Our code release will contain both the implementation of our algorithm and scripts to reproduce all plots.

---

> > > > ### Comment · Reviewer_rPUG · 2022-11-17
> > > > **‘ex-ante’ DP**
> > > >
> > > > 1. I still wonder why Def. 2 is ex-ante as you have used $(\varepsilon(\mathbb{A}, \mathbf{d}), \delta)$-DP in Def. 2 where $\mathbb{A}$ is a particular subset of all outcomes. Only after running the whole program can we know which $\mathbb{A}$ is being used and what $(\varepsilon, \delta)$-DP we can have.
> > > >
> > > > 2. It looks good to me now.
> > > >
> > > > 3. It looks good to me now.

---

> > > > > ### Author Response · Authors · 2022-11-18
> > > > > **More Discussion on 'ex-ante' DP**
> > > > >
> > > > >
> > > > > Thanks for the question. Note that each run of a randomized program only returns a single outcome. If we complete the whole program, then $\mathbb{A}$ becomes a single outcome and Def. 2 indeed degenerates into ex-post DP. However, if we do not complete the whole program, $\mathbb{A}$ could contain many, even infinite, outcomes. In this case, the randomness of the outcome over $\mathbb{A}$ is still a property of distribution (ex-ante). In the case of $T$-step DP-SGD, a single outcome is a tuple that contains the models from step $1$ to $T$ (because of the composition). We fix the randomness in the first $T-1$ models to compute the privacy parameter while the $T_{th}$ model remains a random variable due to the last noise.  In other words, we can get $\mathbb{A}$ and corresponding $(\varepsilon,\delta)$-DP after we run the first $T-1$ steps rather than the whole program.  We have further clarified that Def. 2 is ex-ante when $\mathbb{A}$ contains more than one element, in which case the algorithm output is still a random variable over $\mathbb{A}$.

---

### Author Response · Authors · 2022-11-15
**General Response to All Reviewers**

We thank all the reviewers for their valuable comments and suggestions. We have revised our submission according to the reviews. All notable changes are highlighted in blue color. Below we summarize two main changes.

1. We run Algorithm 1 without individual clipping, and all our conclusions still hold. The results are reported in Appendix C.2. In Section 3, we add more discussion on the use of individual clipping. See our reply to Reviewer rPUG and Reviewer JYd4 for more details.

2. We clarify the objective of our membership inference attack experiments.  They are used to demonstrate that the disparity in privacy still correlates with the unfairness in utility when deep models are trained without DP. We believe this is a natural follow-up after we reveal the correlation for models trained with DP and will be interesting to the community.

---

### Decision · Program_Chairs · 2023-01-20

**Decision:**

Reject

**Justification For Why Not Higher Score:**

N/A

**Justification For Why Not Lower Score:**

N/A

**Metareview: Summary, Strengths And Weaknesses:**

This paper proposes an efficient algorithm to compute privacy loss for individual examples when releasing models trained by DP-SGD. It presents an interesting observation that the training loss and the privacy loss of individual examples are correlated, which implies groups that are underserved in terms of utility are also underserved in terms of privacy guarantee.

Strengths:
It presents an interesting observation that the training loss and the privacy loss of individual examples are correlated.

Weaknesses:
There is a gap between the output-specific individual DP and Theorem 3.1 and the experiment observations.


**Summary Of Ac-Reviewer Meeting:**

The reviewers found that the theoretical part of this paper is incremental, while the experimental part is interesting. There is a gap between the output-specific individual DP and Theorem 3.1 and the experiment observations (i.e., groups that are underserved in terms of utility are also underserved in terms of privacy guarantee).